# ACCU³RATE: A mobile health application rating scale based on user reviews

**Milon Biswas**[1]☯, **Marzia Hoque Tania**[2]☯, **M. Shamim Kaiser**[3]*, **Russell Kabir**[4],
**Mufti Mahmud**[5], **Atika Ahmad Kemal**[6]

1 Computer Science and Engineering, Bangladesh University of Business and Technology, Mirpur, Dhaka, Bangladesh, 2 Department of Engineering Science, Institute of Biomedical Engineering, University of Oxford, Oxford, United Kingdom, 3 Institute of Information Technology, Jahangirnagar University, Savar, Dhaka, Bangladesh, 4 School of Allied Health, Faculty of Health, Education, Medicine and Social Care, Chelmsford, United Kingdom, 5 Department of Computer Science, Nottingham TrentUniversity, Nottingham, United Kingdom, 6 Management and Marketing at Essex Business School (EBS), University of Essex, Colchester, United Kingdom

☯ These authors contributed equally to this work.
* mskaiser@juniv.edu

## Abstract

### Background

Over the last decade, mobile health applications (mHealth App) have evolved exponentially to assess and support our health and well-being.

### Objective

This paper presents an Artificial Intelligence (AI)-enabled mHealth app rating tool, called ACCU³RATE, which takes multidimensional measures such as user star rating, user review and features declared by the developer to generate the rating of an app. However, currently, there is very little conceptual understanding on how user reviews affect app rating from a multi-dimensional perspective. This study applies AI-based text mining technique to develop more comprehensive understanding of user feedback based on several important factors, determining the mHealth app ratings.

### Method

Based on the literature, six variables were identified that influence the mHealth app rating scale. These factors are user star rating, user text review, user interface (UI) design, functionality, security and privacy, and clinical approval. Natural Language Toolkit package is used for interpreting text and to identify the App users' sentiment. Additional considerations were accessibility, protection and privacy, UI design for people living with physical disability. Moreover, the details of clinical approval, if exists, were taken from the developer's statement. Finally, we fused all the inputs using fuzzy logic to calculate the new app rating score.

### Results and conclusions

ACCU³RATE concentrates on heart related Apps found in the play store and App gallery. The findings indicate the efficacy of the proposed method as opposed to the current device

**Data Availability Statement:** All data files used in this study are available from the Dryad database (data link: https://doi.org/10.5061/dryad.jdfn2z3bf).

**Funding:** The author(s) received no specific funding for this work.

**Competing interests:** The authors have declared that no competing interests exist.

scale. This study has implications for both App developers and consumers who are using mHealth Apps to monitor and track their health. The performance evaluation shows that the proposed mHealth scale has shown excellent reliability as well as internal consistency of the scale, and high inter-rater reliability index. It has also been noticed that the fuzzy based rating scale, as in ACCU$^3$RATE, matches more closely to the rating performed by experts.

## Introduction

The world population is rising rapidly which is causing the present healthcare facilities to fall short in meeting the need. To cope to this need, many countries are reforming their healthcare provisioning related to assistive technologies which is a major move aiming to integrate clinical care and consumer health [1, 2]. However, there is a clear imbalance in the supply and demand of healthcare facilities in majority of the countries. This imbalance is not only seen in developing countries, but also in developed ones during high demand scenarios such as the current novel coronavirus disease (COVID-19) pandemic [3, 4]. In addition to insufficient availability of lower-level government healthcare facilities, in a multi-tier healthcare delivery system it's appropriate functioning is severely hampered by inaccessibility to secondary and tertiary healthcare services [5].

In recent years, the massive deployment of the information and communication technology (ICT) infrastructure and enormous improvement of tele-density along with availability of relatively inexpensive smart gadgets, mobile-based therapies (called mHealth) have aimed to confront these issues [6, 7]. These mHealth apps have facilitated medical research and practice in public health through the usage of mobile devices such as tablets and smartphones [8–10]. Paralelly, since the last decade, "patient-centric healthcare" has been gaining popularity in delivering personalised healthcare [11–14]. The technological advances in healthcare provisioning, such as development of mHealth and various portable Health (pHealth) app, allow people to make informed decisions about medical services to monitor their condition, and recover faster from illnesses [15–17]. An mHealth-based patient-centric healthcare approach includes personalised care facilities to meet the demands of the individual needs of a patient. The mHealth app makes it easier for patients to gain quicker access to medical care, enables real-time and continuous patient monitoring, and facilitates medication intake accuracy to increase patient safety [18]. This has been applied to promotes self-management of many chronic illnesses such as diabetes, hypertension and cardiovascular disease [19–22]. This increasing trend in the usage of mHealth apps by consumers has attracted a significant portion of academic research in understanding consumer experience, decision making and behaviour in selecting a pertinent mHealth app over similar alternatives [23–25]. As a result, it's important to understand that mHealth apps are based on evidence and behavioural research in order to encourage long-term use and have a significant impact on health outcomes [10, 26, 27].

Initially, research in the "digital healthcare industry" focused on defining and cataloguing mobile app functionality and characteristics [28, 29]. However, if consumers are expected to use mHealth apps on a consistent basis in order to engage in self-management of their own well-being, it is critical to obtain a deeper understanding of the characteristics and methods that contribute to long-term app usage decision making [10, 30–32]. It is easily noticeable that quite some efforts have been put to address the need of consumers in a healthcare app. Approximately over 100,000 apps are available in the Apple iTunes Store and Google Play Store pertaining to categories such as medical, health and fitness [8]. Recent research shows

that an average smartphone user may have around 80 apps installed on their phone, however, usage of these apps is restricted to only 9 apps per day [33].

The motivation for this paper arises from the very little conceptual understanding that is present on how users' online reviews determine an app's rating, particularly, in the context of mHealth apps. Studies on mHealth apps primarily focus on users' long term set of experiences [10], theorise how cognitive factors impact users' continued intention of using mHealth apps [34] and examine factors influencing the download of these apps [35]. Previous research shows that an app's features and characteristics make positive contributions to the app's rating, thus it is worthwhile to investigate further the rating mechanism of mHealth apps [1]. A set of multidimensional features may better explain a large portion of an app's rating. Hence, it is imperative to gain richer insights into the apps available in stores and quantify the quality of each app's feature through evaluation of user reviews using artificial intelligence (AI)-enabled text mining methods.

Moreover, the limited AI-led research operationalizes users' reviews for assigning the rating of mHealth app. Various dimensions may represent the propensity of a review in attracting users to download a specific app, as user reviews are generally known to be informative and help users finding the most appropriate one. Thus, for a variety of stakeholders (e.g., healthcare providers, app developers and patients) in the healthcare industry, it is pertinent to gain deeper insights on any app's functionality and characteristics from multiple perspectives that have been extracted through text mining of user reviews. Furthermore, AI is an emerging field of research that focuses more on detailed text information generated in reviews. Previous research has used content analysis which is very time consuming, especially when the amount of textual data is very high [1, 35–37]. Text mining has considerable amount of popularity leading to different techniques to quantify textual content [38–41]. While research has evidenced mixed results on the influence of semantic characteristics, other scholars have examined the effects of textual characteristics in online user reviews for helpfulness and effectiveness in online consumer retail markets [40].

However, there is no consistent evidence of how text mining user reviews may distinctly influence mHealth app rating based on multidimensional measures. Thus, it becomes imperative to evaluate the effects of textual characteristics in determining user reviews for mHealth apps. We draw on this knowledge gap that motivates this study and sets it apart from previous studies. We explore the effects of various characteristics of online user reviews and carry out sentiment analysis of mHealth apps which may then assign a new rating, substantially different from the original rating. In the process of creating this new rating, in addition to the existing user inputs (*user star rating*, *user text review*), there are other important factors, such as, *user interface (UI) design*, *functionality*, *security and privacy*, and *clinical approval and certification*, also needs to be taken into consideration. In view of all these factors, this paper presents an AI-enabled mHealth app rating tool, named, ACCU³RATE (clinical Approval & CertifiCation, fUnctionality, Ui degisn, User Rating privAcy & securiTy, and user rEview), based on six factors that influence the mHealth app rating scale. However, considerations such as functionality, security and privacy, UI design and clinical approval details were taken from the developer statement and details provided by the app store.

Within context of the current work, fuzzy logic has been used in determining the relationship among the features. It is highly elastic and is able to incorporate the views of experts into the model in the form of IF-THEN statements. Jumbled, imprecise, inaccurate, and erroneous knowledge can be inputted to the fuzzy system. Thus, a fuzzy logic knowledge fusion technique can be utilized to integrate the effect of all factors mentioned above and determines the new app rating value for the target mHealth app. Our findings suggest that the semantic feature of reviews have a significant impact on the rating score of the mHealth app.

The major contributions from this paper can be summarized as–

- We describe AI-enabled mHealth app rating tool, called ACCU$^3$RATE, which takes multidimensional measures such as user star rating, user review and features declared by the developer to generate the rating of an app.

- We present AI based text mining and sentiment analysis to evaluate the users review for selected apps.

- We introduce a fuzzy logic based knowledge fusion technique that determines the score of the mHealth app based on input factors.

- The results show the effectiveness of selected factors in the rating score of the mHealth app.

The paper is structured as follows. Section provides a theoretical review of mHealth applications and determinants that affect mobile rating. Section describes basics of mHealth app and factors affecting the mHealth app rating scale. In Section, we propose the AI-enabled app rating model. Section shows results and discussions, and finally the work is concluded in Section, followed future direction of this research.

## Literature review

### Overview of mobile health applications

Over the last few years, we have observed an excessive evolution of mobile health applications than any other innovation in health care [42]. The term 'app' is an abbreviation of the word 'application' that refers to a program that has been developed for a specific purpose and is generally configured to run on handheld devices like cell phones, tablets, computers, and certain wearable devices such as smartwatches [43]. In recent times, we have observed a proliferation of mobile apps that are mostly created for only two of the most important tech giants of smartphone operating systems- Apple IOS and Google Android. The World Health Organization (WHO) defines these applications as, 'medical and public health practice supported by mobile devices such as mobile phones, patient monitoring devices, personal digital assistants (PDAs) and other wireless devices'. Similarly, according to Hamel et al. [44] mHealth 'is the use of portable devices such as smartphones and tablets to improve health'. mHealth apps in practice run on mobile devices with remote network connectivity and dynamic execution of contexts [45]. Context is any information that can be used to characterize the situation of an entity, being either a person, place or object [46]. The context awareness system collects real-time data from patients in a comprehensive manner and presents them to health care professionals to manage their tasks in order to increase the quality of patient care [47, 48].

Extant literature on mHealth applications exhibits how mHealth apps are commonly used for health education, disease self-management, remote monitoring of patients, and collection of dietary data [49, 50]. The information collected through mHealth apps can provide useful support for encouraging health behaviour change, chronic disease self-care and effective management of many conditions while keeping healthcare providers informed of the patient's condition [50]. This also strengthens the relationship between the patient and the provider [51]. According to Scoping review findings, mHealth apps provide the potential for general practitioners to take medical history and make diagnoses, perform some physical examinations, aid in clinical decision making, and manage long term disease-specific care and promote general health well-being [52]. Whilst traditionally, health care is delivered to individuals through face to face interaction with health care professionals, technology advancement has facilitated communication between patients and health care professionals [53]. Moreover, mobile phone apps are commonly incorporated into the design of health promotion programs both face-to-face

and online [54]. Based on these, mHealth apps are mainly categorised into two categories—lifestyle management (such as fitness, lifestyle modification, diet and nutrition) and chronic disease management apps (such as mental health, diabetes, and cardiovascular diseases). The other categories include- self-diagnosis, meditational re-minders, and patient portal apps [43].

Studies document how mHealth apps provide information and health behaviour interventions at a meagre cost, that is, easy to access and personalized for user-specific needs [55]. A systematic review of mHealth apps for chronic diseases showed that there are improvements in patient lifestyle related to, healthier eating, weight loss, controlled blood pressure and glycemic levels, including treatment awareness options among patients [21]. With mHealth apps, users can receive important information through push notifications and can monitor their health status at a personal level. Litman and his colleagues [56] from their mobile exercise App research found that exercise app users are more likely to exercise during their leisure time as compared to those who do not use exercise apps. They also highlighted that exercise apps may potentially increase exercise levels and health outcomes such as BMI levels. However, it is also noted that mHealth apps also pose some limitations, and users may face a life-threatening situation if the mHealth apps are not switched to use and made available all the time. In addition, private and sensitive communications are exchanged between the user and the health care provider through these apps. Hence, usability studies should be sufficiently evaluated to determine the practical way in assessing mHealth apps usability [57]. This is critical as any app developer can develop and market a mHealth app through the app store without conducting any usability test [58]. Thus, studies emphasize the importance of a regulatory body to assess the accuracy, quality and performance of health apps, as some mHealth apps were hacked, leading to patient data breaches [43]. Moreover, without any appropriate security measures and processes in place, mHealth app users may become exposed to the severe repercussions of information security and privacy infringements [59]. Hence, we note that there are no general international standards that a data and privacy policy should offer users. This further suggests that, information on how privacy policies determine app star rating is relatively absent. Many privacy policies do not focus on the security features related to the app and therefore do not create confidence in usage for users. In most cases, users may be ignorant and are more likely to use apps which have ambiguous privacy policies [60].

Fig 1(A) shows number of mHealth apps uploaded in the app store each year; Smart phone penetration in the last five years is illustrated in Fig 1(B). The pie chart shows various categories of apps in terms of percentage (See Fig 1(C))

We have found five scales, such as THESIS [42], MARS [61, 62], Brief DISCERN [63], uMARS [64] and ORCHA-24 [65], THESIS [42] is an open-source application rating tool that

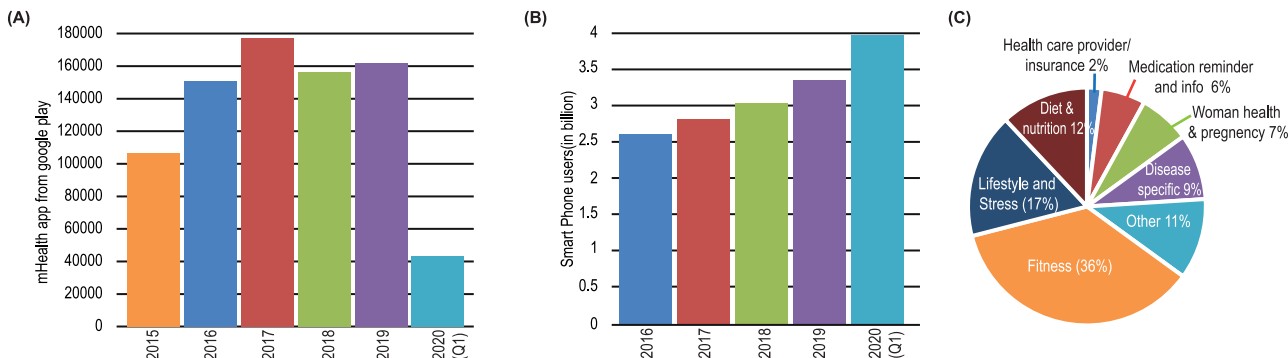

**Fig 1. (A) Year-wise number of mHealth apps uploaded in the google play, and (B) Smart phone penetration in the last five years, and (C) Pie chart shows various categories of apps in percentage.**

is performed using a Delphi process. The tool can calculate usability, risks, and health app benefits. Authors used a standardized method for selecting chronic disease applications with 4 stars and < 4 stars, then scored them with THESIS for their reliability and internal consistency of the tool. The THESIS testing indicates that applications serving chronic illness patients can perform better, particularly in terms of privacy/safety and interoperability. THESIS guarantees further testing and can lead software and policymakers to continue improving app efficiency, enabling applications to enhance patient results more reliably.

Mobile app rating scale (MARS) [66] is an app rating scale for mHealth that has been developed to measure the reliability and efficiency of mHealth applications. The MARS Scale uses four objective quality scales: "functionality", "aesthetics", "engagement" and "information quality"; and one subjective quality scale; which were refined into the "23-item MARS". The MARS has shown excellent intra-class correlation accuracy and reliability.

The Brief DISCERN score [63] can assess quality of mHealth apps for panic disorder (PD) using factors such as transparency, interactivity, self-help score, and evidence-based content. The Brief DISCERN score showed a number of linear regressions that considerably predicted the content quality and self-help performance.

An end-user version of the MARS, called uMARS [64], proposed to assess the quality of mHealth apps. The uMARS had outstanding internal consistency for all subscales with strong individual alphas. The uMARS are used to accumulate app reviews at 1-, 3-, and 6-month intervals. The uMARS had strong individual alphas for all subscales and had outstanding internal stability.

The Organization for the Review of Care and Health Applications- 24 Question Assessment (ORCHA-24) [65] is National Health Service (NHS) approved scale which can evaluate the quality of applications for Chronic Insomnia Disorder on the Android Google Play store for high quality and low-risk health apps. The ORCHA-21 system performed evaluation using evaluation criterion such as efficiency; engagement; protection and privacy; user interface, and quality of clinical efficacy. Table 1 shows the various mHealth app rating scales found in the literature.

None of the rating scales considered, to the best of our knowledge, various characteristics of online user feedback and performed sentiment analysis for mHealth applications, which could then be assigned a new rating, substantially different from the original rating. Thus, the proposed mHealth app rating tool, called ACCU³RATE, uses evaluation criteria such as user star rating, user text review, UI design, functionality, security and privacy, and clinical approval.

**Table 1. Various mHealth app rating scale.**

| Ref. | Scale Name | Factors | Performance Metric | Results |
|------|------------|---------|--------------------|---------|
| [42] | THESIS | U; S/P; THC; T | IRR | $k > 0.31$ |
|      |          |                | ICC | $\rho_T = 0.85$ |
| [66] | MARS | E; F; Es; I; Q | ICC | $\rho_T = 0.79$ |
| [63] | Brief DISCERN | T; Q | LR | $R^2 \approx 0.7$ |
| [64] | uMARS | E; F; Es; I; Q | ICC | $\rho_T = 0.90$ |
| [65] | ORCHA-24 | Dg; Cea; UX | IRR | $k > 0.88$ |
|      |          |              | ICC | $0.84 \leq \rho_T \leq 0.93$ |

*In Factors Column*: U–Usability; S/P–Security/privacy; THC–Technical and Health Content; T–Transparency; E–Engagement; F–Functionality; Es–Esthetics; I–Information; Q–Quality; Dg–Data governance; Cea–Clinical efficacy and assurance; UX–User experience and engagement;

*In Performance Metric Column*: IRR–Interrater reliability; ICC–Intra-class Co-relation Co-efficient; LR–Linear Regression;

## Determinants of mobile health app ratings

At present, there is little known about which mHealth apps are most commonly used and whether they align with app quality [67]. Majority of the mHealth apps found in the app stores are available in trial versions to attract potential users or as a promotional strategy to increase the adoption rate [68]. Fiordelli et al. [9] in their systematic review brought to our attention that there are millions of mHealth apps (free and paid) publicly available on app stores, and to-date they have not been evaluated them. Byambasuren et al. [58] highlighted that low quality and lack of effectiveness hinders the usability of mHealth apps, as the majority are developed by non-health care organisations which raises questions about the accuracy and trustworthiness of these apps [69].

While evaluating behavioral mHealth apps, studies report that app ratings are broadly inconsistent and often contradictory while most popular behavioural health apps rating are not favorable [67]. Furthermore, other frequently downloaded behavioral mHealth apps have questionable support in the literature and offer no evidence-based behavioural support. For example, research on the usage of mHealth apps by cancer survivor patients revealed that some apps used theoretical models of behaviour change, but the majority of the apps in the market did not apply any of those theoretical elements in their apps [55].

Some scholars suggest that independent and reliable sources should evaluate mHealth apps and should recommend a collection of trustworthy apps to health care professionals to, be referred to patients [58]. According to Levine et al. [42] an app rating tool can be developed to identify mHealth apps that may cause potential harm and breach the privacy policy. When an appropriate mHealth app is prescribed by health care professionals, they must be confident about the functionality of the app, maintenance of data privacy and its usability [58]. A survey conducted among mHealth app users in the USA found that 82 percent users would change providers if they were aware of alternative apps which were more secure [70]. There is also no clear evidence available to suggest that an app developed by a health care organization will have more usability. Bivji et al. [69] identified that price, user ratings, in-app purchase options and in-app advertisements were the greatest predictors of app downloads from the app store.

Other studies present that the actual usability and quality rating scales are primarily developed only for professionals, and not for end-users who are patients or care providers. It is widely recognized that rating scales that are usable by all end users would make mHealth apps more accessible and meaningful to consumers [71]. Janatkhah et al. [72] found that a mHealth app's usability is associated with the education level, employment and place of user's residence. While there is evidence of a weak relationship between user app ratings and usability and clinical utility of the app [73] in the literature, generally we note that users make decisions related to app use, based on the title, price, star ratings, reviews and number of downloads [74]. Furthermore, mHealth users disproportionally favour apps that take advantage of unique features of smartphones with higher ratings and present the real benefits to users to keep track of their health records [45].

Generally, we observe that mHealth app users' opinions or satisfaction levels, star rating system, app description of checklists are some common features used to measure the quality of the apps. However, scholars argue that none of these methods are scientifically adequate to measure the quality of the apps [75]. Singh et al. [73] identified that store rating alone is not enough to determine whether an app is of high quality but inclusion of clinicians and users' reviews will ensure that the recommended apps are usable and clinically useful.

Further, we note that in the literature users' reviews of mHealth apps in relation to app rating is generally scarce. There is researchs on the effect of online assessment (i.e. whether reviews in an examination collection are overwhelmingly positive or negative) on different

dependent variables, in particular on the perceived utility of the reviews and attitudes towards a product [76–78]. This operationalization of review effectiveness warrants helpfulness and facilitates user's decision-making [79] whilst inspiring trust and confidence among other potential users [80–83]. Nevertheless, the existence of a contradictory total rating reduces the credibility of the analysis due to its negative impact on consumer product attributes [84].

Some scholars are sceptical about the review 'helpfulness' level related to user votes, and suggest other diverse features, such as basic, stylistic, and semantic characteristics by applying data mining techniques from online user reviews [40]. Although this multi-process conceptualization of multiple determinants for review effectiveness is emerging in the literature, some scholars are of the view that participation in reviews and polarity of reviews may be driven by different consumer behaviors [79] and emotions [85] so should be studied separately. Although it is observed that positive review sentiments increase readership [86].

The basis for digital health research is often primarily based on cataloging the number and types of apps dealing with a specific health issue that assess the theoretical effect of such features on user experience or identifies the behavioral concepts in a subset of mHealth apps. Studies can also explore a variety of multi-dimensional app features that contribute to a better user experience and eventually alter long-term behaviours [1]. Based on these gaps from the mHealth app rating, our study aims to extend current knowledge on how users' perceptions and app reviews can generate new app rating that may be different than the original/existing app rating. This paper also presents a new methodology, an AI-enabled mHealth app rating scale, which takes multidimensional measures to generate app rating. By adopting text mining as a novel methodology to extract various semantic characteristics from reviewers' texts, this paper evaluates user review determinants on mHealth app ratings.

## mHealth app quality rating scale

### mHealth applications

The mHealth applications are an important part of the forthcoming next-generation Internet of Healthcare Things (IoHT) [87]. This paper categorises the mHealth apps into the following six types.

**Wellness management app.** The Wellness management app tracks health aspects such as walking, sleeping, weight, body mass index, temperature, blood pressure, oxygen saturation (oxygen), pulse, consumption of food and water, etc. These apps allow users to monitor aspects of their health and lifestyle.

**Disease management app.** Aging, demographics, lifestyles, and chronic diseases have been adding an increasing amount of pressure on health care systems around the globe. In order to reduce such pressure, disease management software is employed to assist the healthcare system, aiding in monitoring signs/symptoms, tracking medications' intake, and other side effects.

**Educational app.** Educational applications offer disease-focused education and self-management training.

**Self-diagnosis and decision-making app.** An emergency patient assessment with a symptom management questionnaire is performed via the self-diagnosis and decision-making app.

**Digital therapeutics and rehabilitation.** Digital therapeutics and rehabilitation may be used to prevent, control, or treat disease.

**Behavioral change and medication.** Behavioral improvement and medication mHealth apps identify behavioral changes that improve patients health and facilitate personalized medicine (see Fig 2).

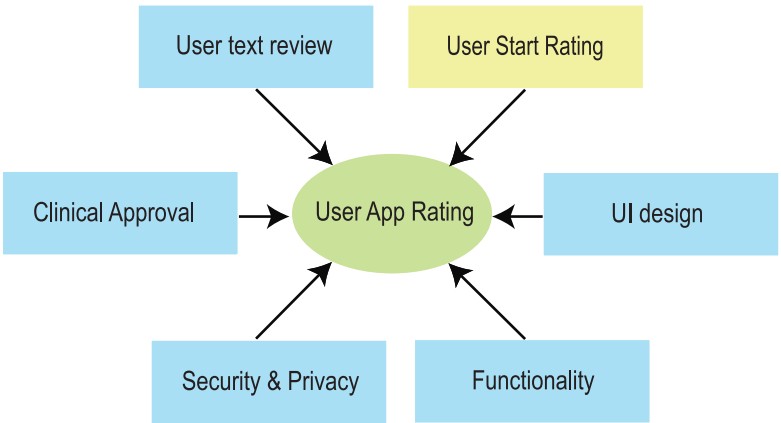

**Fig 2. Factors affecting mobile app review.**

## Factors affecting mHealth app rating

In determining mHealth app quality scores important factors related to the app such as the clinical approval, UI design, functionality, security and privacy are often ignored by the current practice. The complete list of factors which are considered essential in determining an appropriate rating of the mHealth app are provided below:

**Approval and certification.** Apps, intended for disease diagnosis, cure, mitigation, treatment, or prevention of disease, can be considered as medical devices under the Food and Drug Administration (FDA) regulation [88]. Such an app also has the ability to affect the structure or any function of the human body.

Under the latest guidance of European Medical Device Regulation (EU MDR) [89], apps, having a medical purpose such as prevention or diagnosis of a disease, treatment or alleviation; compensation for an injury or disability; investigation, replacement or modification of anatomy or physiological processes; control of conception; and prediction and prognosis require regulatory approval.

Medical approval is not mandatory for all mHealth applications, as many applications do not engage in clinical diagnosis or intervention [90]. For example, the Wellness Management Apps do not require clinical approval [91].

The mHealth app is designed and developed to meet a specific purpose. The developer offers criteria, specifications, and recommendations after planning the app. Standardization ensures that the facilities, protection, reliability, and consistency of a newly developed mHealth app match the objective. A standardization board must approve all mHealth apps as the mHealth apps need to be budget free and should deliver the expected performance. Authorization and certification provide credibility to a mHealth app and help a physician choose an app for the patient.

**Functionality.** The app designer's first aim is to create an mHealth app with full functionality and a high-quality experience. Having too many features in an app on a small screen can cause distraction, and it may not be easy for the app user to get all the essential information. The selection of key features is, therefore, essential to ensure consistency of the experience. The well-designed mHealth app must provide expert opinions, meet all problem domain criteria, and provide better accuracy, accompanied by the HCL law [92]. The optimal functioning of the mHealth app can involve alert/ warning, connectivity via multiple interfaces (such as NFC, LiFi, WiFi, Bluetooth, etc.) of a mobile, well documented, stable output, customizable,

low-cost/free, family-friendly, easy-to-use, recording for viewing recorded data. The easy-to-use measure refers to either a qualitative/quantitative assessment of how quickly targeted users use the mHealth app.

**UI design.** The design of the user interface (UI) is one of the key concerns of the app developer and depends on hardware and technical improvements (such as touch screen / appearance), aesthetics, user context (such as history, age and disability) and functionality. Some of the challenges are limited screen size and a high demand for visual attention. The app designer therefore aims, to provide a sleek interface with important features to ensure the quality of experience.

**Security and privacy.** Security and privacy are other critical concerns for mHealth. Privacy protects a patient's name, age, sex, cultural, mental, physical/physiological, socioeconomic, and medical factors. A user permission is also required once the third-party agency receives the personal information transfer app, and any unnecessary data must be deleted. In order to maintain privacy, security, and confidentiality, the app developer must choose the appropriate access control and encryption protocol using fewer mobile device resources.

**User star rating.** User star ratings, also called user star rankings, are average values based on user input on Apps and range from 0 to 5. They are generally averaged over a period of time or by the number of times the App has been updated.

**User review.** User text reviews provide insight into current trends and problems encountered when using an app. Because of the personalized feedback, the app developer will be able to construct apps based on the user experience and expectation. Thus user text reviews is an important factor to be considered and the normalzied value range from 0 to 5.

## Methods: AI-enabled text data mining

### ACCU³RATE app rating model

The proposed app rating tool, called ACCU³RATE for mHealth Apps, provides a multidimensional application quality evaluation using criteria such as user star rating, user text review, UI design, functionality, security and privacy, and clinical approval. The app rating tool eliminates the aforementioned variables from the app developer and generates an improved rating based on the user experience. In the conventional rating, the quality of the app is judged only by user stars. However, our research suggests that the declaration of the developer as well as user text review must be incorporated. The relationship between input parameters and the model parameters is shown in Fig 3.

### Developers declaration

An app developer declares the mHealth app criteria such as clinical approval, standardization certification, functionality, UI design, security and privacy. The app developer's response calculates the initial rating (initial $\in [0, 5]$). Algorithm 1 reflects the complete procedure from calculating the initial rating to changing the rating based on user feedback. At the time of the publication of Apps, the initial rating for an app is created. Based on the user experience (star rating and review), this rating changes. The feature "users recalculation" returns a positive or negative value depending on the polarity of user comments in the Algorithm 1.

**Algorithm 1**: Initial rating based on developer's feedback

```
Result: R_Base
total_topic, R_Base = 0, R[total_topic], counter[total_topic]
if FDA_Approval_Req_Dev_Response() == True then
  total_topic = 5;
else
```

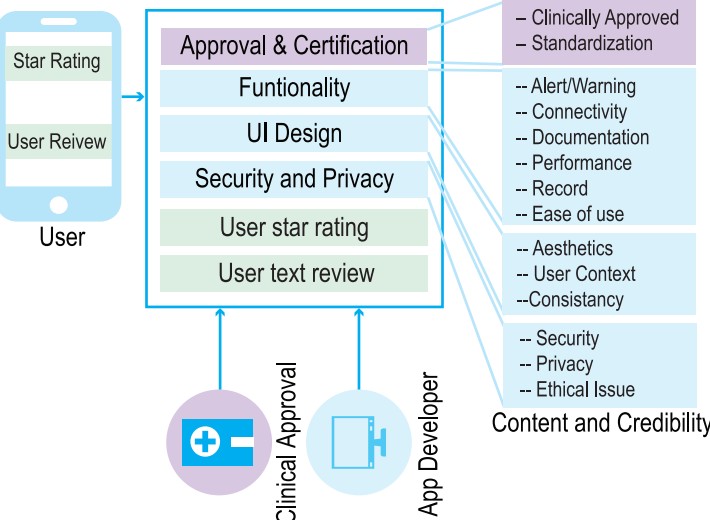

**Fig 3. Relationship between input parameters and model parameters.**

```
  total_topic = 4;
while r = 1 to total_topic do
  R[r] = developer_ans_on_topic_r();          // Initial Rating based on
developer response, 0 or (5/total_topic)
  R_Base += R[r];
end
Publish(R_Base);
```

## User rating and review

The key variables considered in this paper are discussed as follows.

**Star Rating.**   The app's star rating is the base goodness metric, which is determined by the cumulative user input rating ranging from 0 to 5. The user considers it as being trustworthy over personal recommendations while downloading an app.

**Text mining and sentiment analysis.**   The polarity of user reviews in the mHealth app can be extracted using the following steps (see Fig 4).

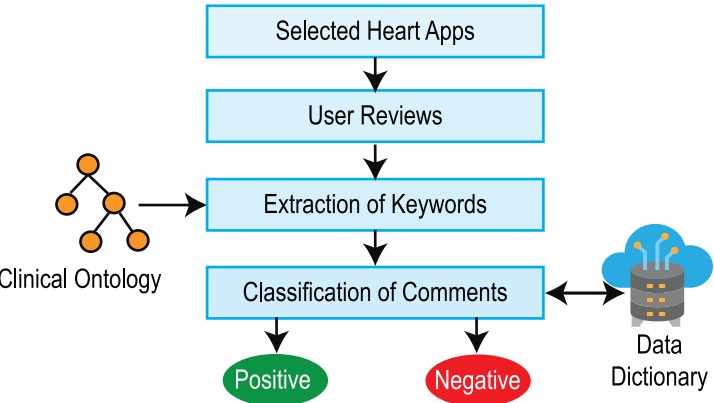

**Fig 4. Flow diagram for opinion mining.**

**Step 1: Selection of mHealth apps**: The mHealth applications are selected from the Google Play and App Store for a specific category (such as health-related apps).

**Step 2: Collection of comments**: Based on the assumption that the user star rating, user review, number of downloaded features, and app features can influence the app's reputation, the aforementioned data for the selected apps is accumulated. NLTK (Natural Language Toolkit) [93], a versatile package of natural language processing algorithms, analyzes the user's critique/comments. All comments from the app review segment are gathered using scrubbing software.

**Step 3: Extraction of Keywords**: Keywords are extracted from a customer's review that contributes to the app's credibility. We break down the analysis of comments into small bits like words and use tokenization techniques, where word tokenization splits the entire text into words. The frequency that relates to the number of each word in the method is then transmitted and counted. In text data, mining stopwords (i.e., am, are, this, a, an, etc.) is viewed as noise. To delete all of these, we created a stopword list and filtered the tokenized word list with it. After tokenization, the words go through stemming, i.e. a normalization process where the term is shortened to the root word or a derived affix is trimmed.

**Step 4: Identification of polarity**: The polarity of the comments is defined based on sentiment analysis [94]. We use the clinical ontology and lexicon-based approach to evaluate emotions from a written text to count positive and negative terms among the review comments. The classification of comments is one of the most important aspects in the text mining industry. We created a data dictionary to evaluate our dataset and, with the aid of the **word2vec** algorithm [95], and then we fit the data dictionary with the comment data. Using the NLTK text corpora and **word2vec** algorithm, these review comments are analyzed to find any constructive negative review comments.

We then use the Naive Bayes classifier [96, 97] to evaluate the positive and negative polarities of the comments. We use two environments, NLTK [98] and Spider [99] for sentiment analysis. Word charts in Fig 5 shows the wordcloud for positive and negative polarities of words found in the comments of the app.

## Approval and certification

mHealth technologies originated mainly from non-medical practice. As a consequence, such applications can lead to poor content, poor treatment and unfavorable outcomes. Inappropriate use of apps by patients and clinicians could lead to a worsening of health conditions if there is no evidence-based guidelines for app growth. Thus, clinical approval is necessary for mHealth applications especially if the app is used for monitoring of clinical data, provide diagnosis, or intervention for a medical condition.

Strict protocols and criteria are essential for ensuring the quality of an app. The mHealth software developer should have an understanding of the multidisciplinary standards such as ISO 14971:2007 [100] and ISO 14971:2019 [101]. Any mHealth app with clinical approval and ISO certification improves the credibility of the app.

Fig 6 shows calculation of "Clinical Approval" value using four questionnaires such as

- **Q1**: Is the app monitored or analyzed patient data or patient specific medical device data/function?

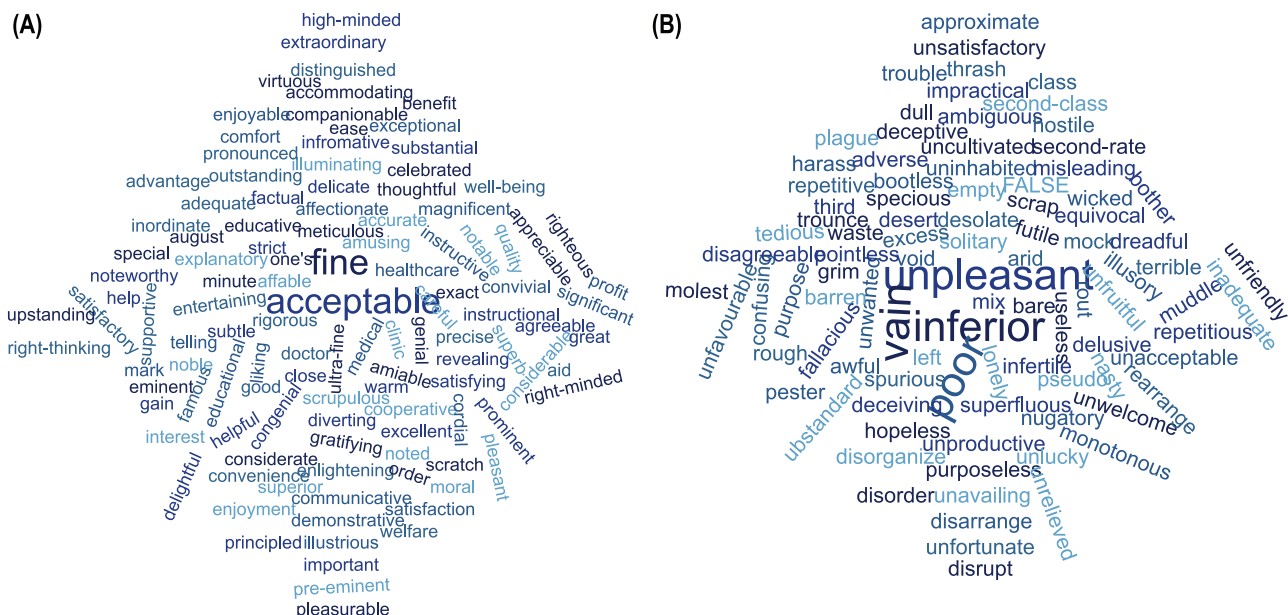

**Fig 5. Word cloud for Polarity of words found in the comments of selected apps– (A) positive polarity words, and (B) negative polarity words.**

- **Q2**: Is the app connected to control the operation, function or energy source of a medical devices?

- **Q3**: Is the app connected to medical device that assists licensed practitioner to diagnose/treat a medical condition?

- **Q4**: Is the app followed any International Standard and received Certificate?

### Fuzzy logic for fusion

The proposed App rating tool ACCU³RATE uses a fuzzy logic controller (FLC) which generates a rating score using criteria such as user star rating, user text review, UI design, functionality, security and privacy, and clinical approval. The FLC is a rule-based system used extensively to design decision-support systems [102]. Fig 7 shows the FLC which combines the knowledge extracted from the user star rating, user text review, clinical approval, UI design,

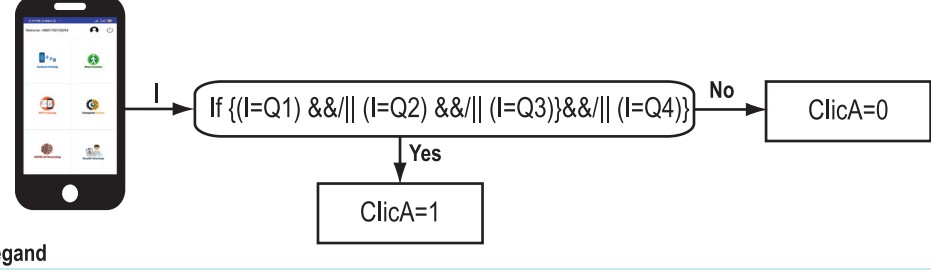

**Legend**

Q1: Is the app monitored or analyzed patient data or patient-specific medical device data/function?
Q2: Is the app connected to control the operation, function or energy source of a medical device?
Q3: Is the app connected to medical device that assists licensed practitioner to diagnose/treat a medical condition?
Q4: Is the app followed any International Standard and received Certificate?

**Fig 6. Calculation of clinical approval value.**

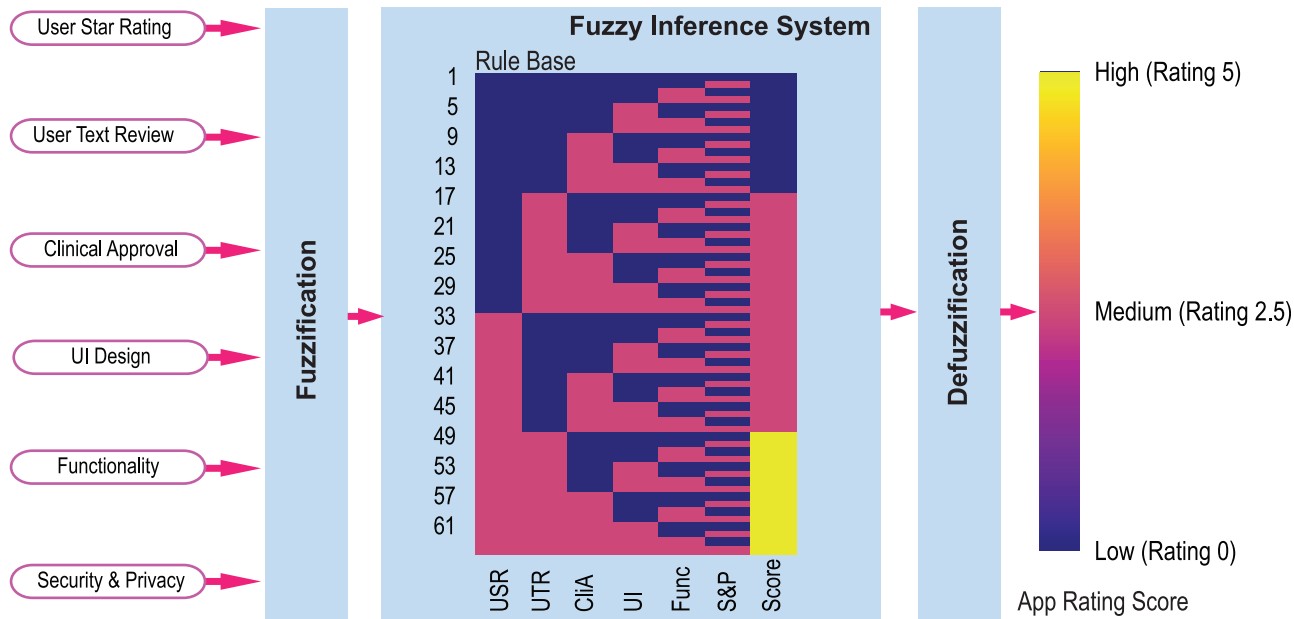

**Fig 7. Fuzzy logic based fusion technique which combines the knowledge extracted from the users' star rating, users' text review, clinical approval, UI design, functionality and security & privacy and thereby generates a score.**

functionality, security and privacy, and thereby generates an App rating. The FLC has mainly three layers; these are fuzzification, rule base and inference system and defuzzification [103].

The fuzzification layer converts the input variables into fuzzy forms using linguistic terms and Gaussian membership functions that convert the crisp inputs into fuzzy inputs. We have chosen three linguistic terms (Low, Medium, and High) for Users' Star Rating (USR); and two linguistic terms (Low and High) for users' text review (UTR), clinical approval (CliA), UI design (UI), functionality (Func), and Security and Privacy (S&P).

The rule layer includes 62 "IF . . . THEN" rules for the system and a fast forward rule selection algorithm is used to select optimal rules for improving the FLC's accuracy and computation. All the rules are represented by a heatmap in Fig 7

After the fuzzy inferencing, all the consequents are integrated by multiplying a weight value. The defuzzification layer converts the fuzzy value into an average value. In this case, the center of the area, the defuzzification method is employed for its simplicity.

Fig 7 also shows the input and output membership functions, and the surf plots refer to the relationship between the input and output variables. Fig 8 shows the input membership and the relationship between the input-output in the rule base system.

## Tool and rating correlation

*Cronbach alpha* is used to measure interrelated items to estimate quality score (i.e., scale reliability) and internal consistency of the scale. On the other hand, the interrater reliability index, called *Cohen's kappa*, determines the extent to which two raters's ratings agree or disagree with each other for qualitative objects.

## Results and discussion

### Data collection

The data collection phase started with selecting appropriate apps to provide the proof-of-concept. The apps were searched using the keywords *heart disease*, *heart related medical app*,

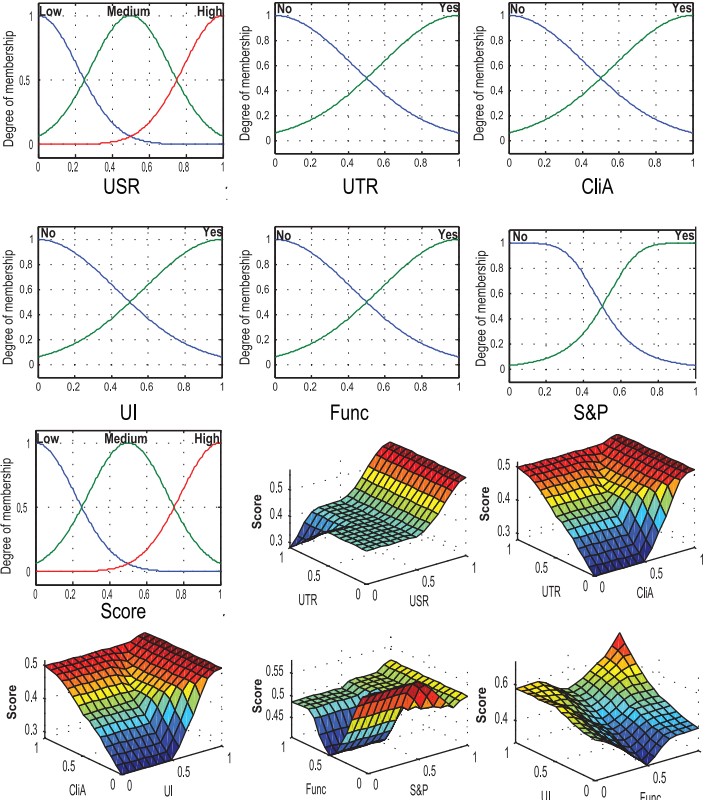

**Fig 8. Input membership and relation between the input-output in the rule base system.**

*healthy heart*, and *heart care* in the Google Play Store and Apple store to evaluate the proposed AI-based app rating scale. A total of 317 applications in different categories have been identified based on the search criteria. Each app was initially qualitatively assessed and screened with titles, descriptions, and associated screenshots provided in the Play Store. Out of the total 317 m-Heath apps, 278 are from the Google Play Store while, 39 are from the Apple Store. Then, we set some exclusion criteria for apps, such as duplicate apps, apps not related to heart disease, apps downloaded less than 5000 times, game and simulation apps, apps not updated in more than 12 months, and non-English apps. Consequently, after filtering based on all these criteria, we selected 43 apps (23 apps from Google Play Store and 20 apps from the Apple Store) for further analysis. Fig 9 shows the case study design and app selection flowchart.

## Correlation analysis

The proposed mHealth app rating tool provides a multidimensional assessment of the App interaction, functionality, aesthetics, subjective quality, security and privacy.

Fig 10 illustrates the correlation coefficient of selected parameters such as user experience (user star rating and text review), approval and certification, UI design, security and privacy. The correlation coefficient ($r$) between the variable is considered for the app scale assessment. User experience has found to have influence on functionality ($r = 0.5$), privacy and security ($r = 0.2$). The UI design also affects star rating ($r = 0.1$) and text review ($r = 0.1$). App functionality is correlated with user star rating ($r = 0.5$) and text review ($r = 0.5$). The fuzzy logic based mHealth app evaluation framework has been employed to produce app ratings based on user star rating, user text review, clinical approval, UI design, functionality, and security and

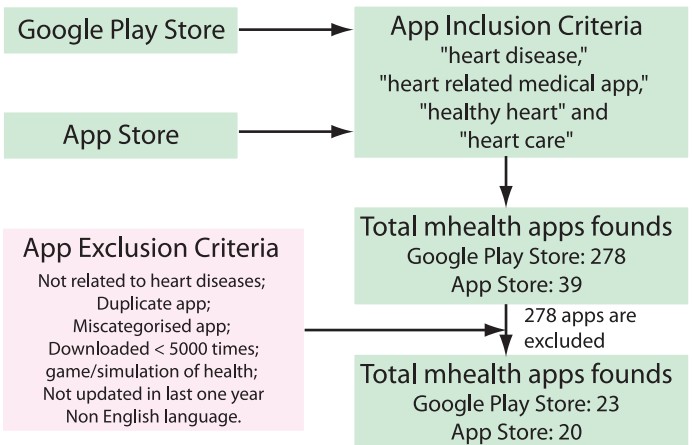

**Fig 9. Case study design and app selection flowchart.**

privacy. From the initially selected 317 apps, our filtering categories (such as consistency, user context, aesthetics, ethical issues, privacy, security, connectivity, alertness, ease of use, record and clinical approval) informed the final selection of 43 apps. The input factors, proposed in this paper, include various model parameters such as users' star rating, users' text review, clinical approval, UI design, functionality, and security and privacy (See Fig 3).

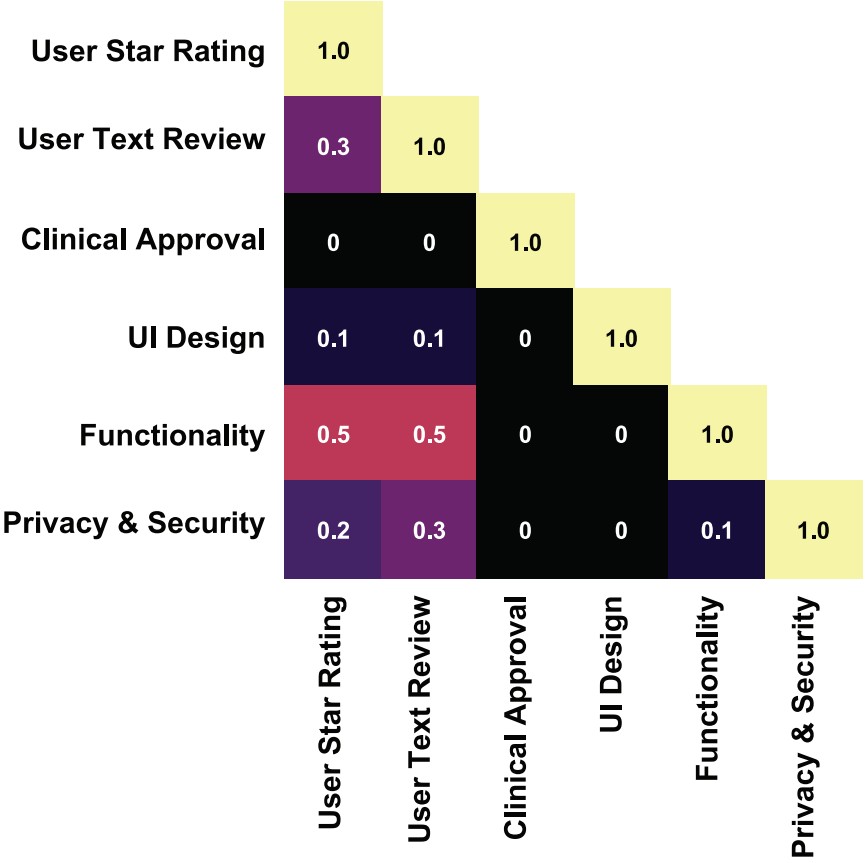

**Fig 10. Correlation matrix illustrating the coefficient among factors considered for the assessment of the app scale.**

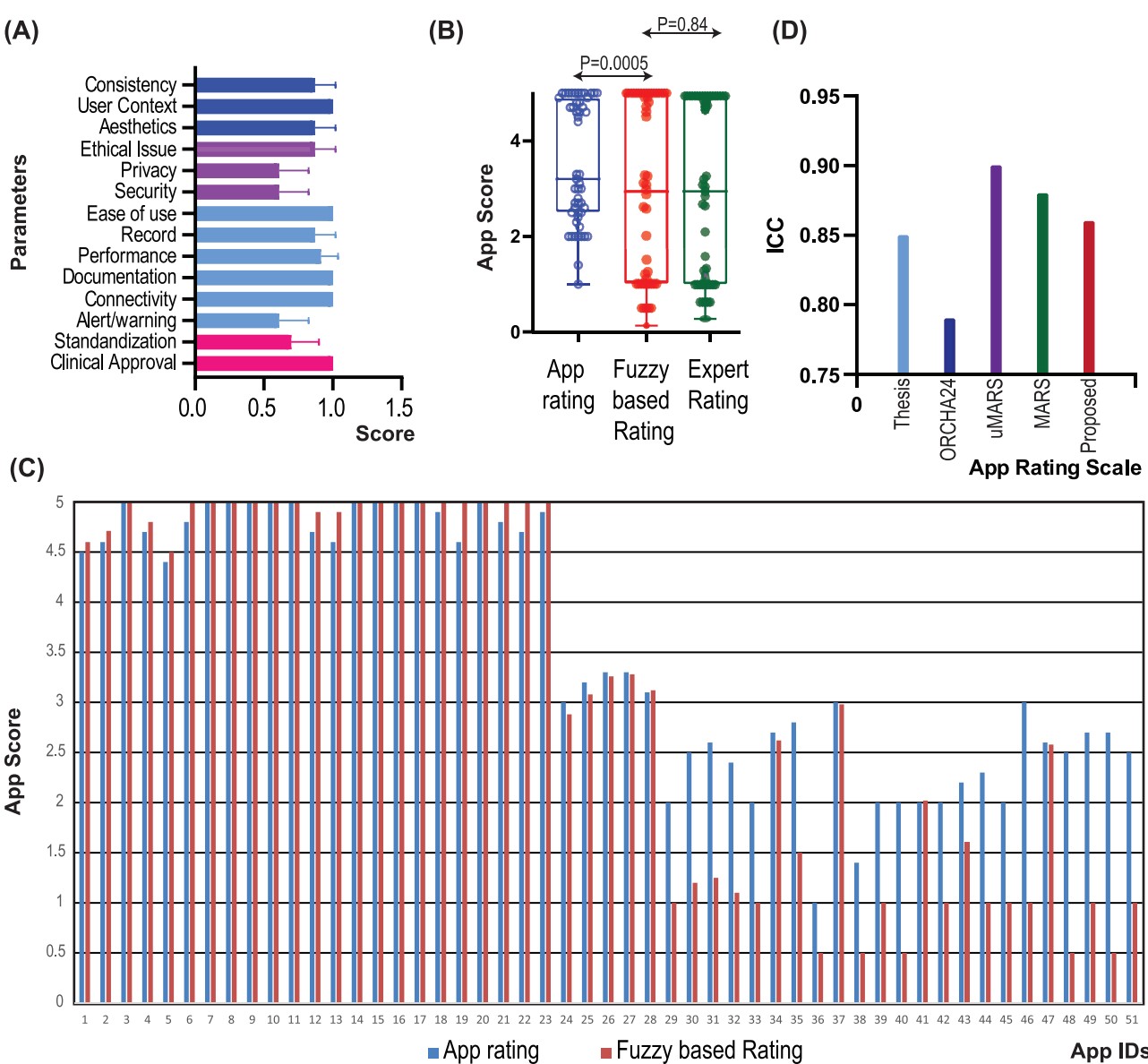

**Fig 11. Comparison of traditional mobile app ratings and fuzzy based app ratings.** (A) Average scores with respect to each parameter for all the selected 43 apps, (B) Box plot for the average rated value for traditional, fuzzy based ratings, as well as ratings based on expert opinion, (C) app scores for traditional and fuzzy based ratings for all the selected apps, and (D) Comparison of various app scales in terms of ICC.

Fig 11(A) shows the average score for the different parameters, while, Fig 11(B) presents the box plot for the average rated value for conventional app rating scores, fuzzy based scores and app scores based on expert opinion. It has been found that the fuzzy based rating has a high variance compared to the conventional app rating, whereas the fuzzy based rating shows a high relationship in contrast to scoring based on expert opinion ($p = 0.84$). Fig 11(C) demonstrates app scores for conventional and fuzzy-based ratings for all the apps chosen. It has been noted that fuzzy based rating decreases the rating value if the selected input parameters have not been found in these 43 apps.

It has been found that the *Cohen's kappa* value varied as: approval and certification (0.44), functionality (0.41), UI design (0.50), security and privacy (0.55). On the other hand, the

*Cronbach alpha* value is found to be 0.86 which shows the ACCU³RATE has an acceptable degree of reliability of scale and internal consistency.

When the proposed app rating scale ACCU³RATE is compared with the THESIS, ORCHA-24, uMARS, and MARS in terms of the ICC on 43 apps (23 apps from the Google Play Store and 20 apps from the Apple Store), it is found that the factors in the same domain have 2-way mixed ICC = .86 (95% CI 0.81–0.92) in contrast to the app scales THESIS: 0.85; ORCHA-24: 0.79; uMARS: 0.90; MARS: 0.88 (See Fig 11(D))

The high value of ICC (ICC = .86) shows components under the same criteria in the proposed app scale resemble one another very closely, implying that the ACCU³RATE app scale is very reliable.

## Conclusion

This article describes a multidimensional mHealth App rating scale that uses user star ratings, user reviews, and developer-declared features to build an app rating. However, there is currently a dearth of conceptual knowledge on how user reviews effect App ratings on a multidimensional level. This study employs an AI-based text mining technique to gain a more thorough knowledge of user inputs based on a variety of parameters. Six variables were identified as influencing factors for the mHealth app rating scale. These factors are– user star rating, user text review, UI design, functionality, security and privacy, and clinical approval & certification and lays the basis of the proposed mHealth app rating tool named ACCU³RATE. The user sentiment were extracted from their reviews and clinical approval facts, if present, were extracted from the developer statement using the Natural Language Toolkit package. Finally, fuzzy logic was used to combine all of the data and produce the rating score. Several heart-related apps available in the Google Play Store and Apple app store were used as a case research to assess the effectiveness of the app rating scale. The ACCU³RATE mHealth scale has shown excellent reliability as well as internal consistency of the scale, and a high interrater reliability index (Cronbach alpha = 0.86). It has also been found that ratings provided by ACCU³RATE has a high similarity score ($p$ = 0.84) to manual scoring performed by experts. The paper has practical implications for App developers who may draw on this knowledge to improve their Apps functionality. Even for mHealth users, new insights are provided on mHealth usability that may help reduce the burdens of healthcare provision through empowering users to manage their own health using innovative mHealth technologies.

## Future work

The immediate next step of our research will be to expand the scale, deploying the presented method for a wider range of applications beyond heart disease to verify the generability of the findings. With the better understanding of deep learning (DL) algorithms [97, 104] and increasing demand for explainability and causability in the AI/DL model by clinicians while not compromising on performance, called eXplainable AI (XAI) [105, 106], the proposed model can be extended by considering ML/DL approaches to design a new mHealth apps rating scale which has the Influence of User Reviews.

## Supporting information

**S1 Data.**
(TXT)

## Author Contributions

**Conceptualization:** M. Shamim Kaiser, Atika Ahmad Kemal.

**Data curation:** M. Shamim Kaiser, Russell Kabir, Mufti Mahmud, Atika Ahmad Kemal.

**Formal analysis:** Milon Biswas, M. Shamim Kaiser, Russell Kabir, Atika Ahmad Kemal.

**Investigation:** Marzia Hoque Tania, Mufti Mahmud.

**Methodology:** M. Shamim Kaiser, Mufti Mahmud.

**Project administration:** Marzia Hoque Tania.

**Supervision:** M. Shamim Kaiser.

**Writing – original draft:** Milon Biswas, Marzia Hoque Tania, M. Shamim Kaiser, Russell Kabir, Atika Ahmad Kemal.

**Writing – review & editing:** M. Shamim Kaiser, Russell Kabir, Mufti Mahmud, Atika Ahmad Kemal.

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
