## [Decision Letter · Decision Letter 0]

6 May 2021

PONE-D-21-09300

The Influence of User Reviews on Mobile Health Application Rating

PLOS ONE

Dear Dr. Kabir,

Thank you for submitting your manuscript to PLOS ONE. After careful consideration, we feel that it has merit but does not fully meet PLOS ONE’s publication criteria as it currently stands. Therefore, we invite you to submit a revised version of the manuscript that addresses the points raised during the review process.

We look forward to receiving your revised manuscript.

Kind regards,

Junaid Rashid, Ph.D

Academic Editor

PLOS ONE

Journal Requirements:

Please amend your list of authors on the manuscript to ensure that each author is linked to an affiliation. Authors’ affiliations should reflect the institution where the work was done (if authors moved subsequently, you can also list the new affiliation stating “current affiliation:….” as necessary).

Please amend either the abstract on the online submission form (via Edit Submission) or the abstract in the manuscript so that they are identical.

Editor Comments :

The idea of the paper is interesting. There are some major concern of reviewer 2 and 3. The paper need improvement. The paper also contain some grammatical and spelling mistakes. Proofreading of paper is required.  

Reviewers' comments:

Reviewer's Responses to Questions

**Comments to the Author**

1. Is the manuscript technically sound, and do the data support the conclusions?

Reviewer #1: Yes

Reviewer #2: Yes

Reviewer #3: Yes

2. Has the statistical analysis been performed appropriately and rigorously? 

Reviewer #1: Yes

Reviewer #2: Yes

Reviewer #3: Yes

3. Have the authors made all data underlying the findings in their manuscript fully available?

Reviewer #1: Yes

Reviewer #2: Yes

Reviewer #3: No

4. Is the manuscript presented in an intelligible fashion and written in standard English?

Reviewer #1: Yes

Reviewer #2: Yes

Reviewer #3: Yes

5. Review Comments to the Author

Reviewer #1: This paper presents an Artificial Intelligence (AI)-enabledmHealth app rating tool

which takes multidimensional measures such as starrating, user’s review and features

declared by the developer to generate apprating.

The paper is well written and can be accepted for publication.

Reviewer #2: This study applies AI text mining technique to develop more comprehensive understanding of users' feedback based on an array of factors, determining the mHealth app ratings.

Overall the paper is well written and easy to follow.

There are some minor issues to be addressed:

- main limitations need to be also explained/discussed

- why a fuzzy logic has been employed? further motivations are needed

- future works are limited. The author should further discuss this section. In the future,

explainable AI aspects should be addressed in order to give a deeper understanding of the proposed

system. In this sense the authors can refer to: A novel explainable machine learning approach for EEG-based brain-computer interface systems; Towards multi-modal causability with Graph Neural Networks enabling information fusion for explainable AI

Generally, this work is important and with the modifications this can be a really useful archival work for the international community. Congrats to the authors and this reviewer hopes that the suggestions help to improve the paper.

Reviewer #3: This manuscript presents a novelty ranking of mobile health applications. This ranking is based on the user's star ranking, users' text review, clinical approval, UI design, functionality, security, and privacy. Results suggest a competitive performance of the proposed ranking compared against app and expert rating. However, some points must be attending.

1. Review some misspelling words (invesgitation on page 2). I recommend reading the entire document to avoid this writing mistake.

2. Algorithm 1 is located several pages below its mention. I suggest including it below of its reference.

3. Into the manuscript are mentioned Questionaries Q1-Q4 for Clinical Approval measure. Please, include a detailed comment about this because it looks like a disconnected element. I understand that they are used for getting the clinical approval value, but the questionaries are not described.

4. Are mentioned five scales for ranking apps: THESIS, MARS, Brief DISCERN, uMARS, and ORCHA-24. Compared with these, how is your proposal? is better, worst, or similar?. Please provides a comparison with these methods.

6. PLOS authors have the option to publish the peer review history of their article (what does this mean?). If published, this will include your full peer review and any attached files.

Reviewer #1: No

Reviewer #2: No

Reviewer #3: No

---

## [Author Response · Author response to Decision Letter 0]

3 Jul 2021

PONE-D-21-09300: The Influence of User Reviews on Mobile Health Application Rating

Editor Comments:

The idea of the paper is interesting. There are some major concern of reviewer 2 and 3. The paper need improvement. The paper also contain some grammatical and spelling mistakes. Proofreading of paper is required. 

Author response: Thank you. We have addressed all the concerns of reviewers. In addition, the manuscript has been carefully proofread to eliminate grammatical and spelling mistakes.

Reviewer # 1

------------------

Response to the Reviewers’ Concerns Reviewer # 1

Comment # 1: 

This paper presents an Artificial Intelligence (AI)-enabled Health app rating tool which takes multidimensional measures such as star rating, user’s review and features declared by the developer to generate app rating. The paper is well written and can be accepted for publication.

Author response: Thank you so much for your encouragement. It is appreciated. 

Reviewer # 2

-------------

Response to the Reviewers’ Concerns Reviewer # 2

Comment # 1: 

This study applies AI text mining technique to develop more comprehensive understanding of users' feedback based on an array of factors, determining the mHealth app ratings. Overall, the paper is well written and easy to follow. 

There are some minor issues to be addressed: 

Concern # 1:

Main limitations need to be also explained/discussed

Author response: The manuscript has been updated by including the scopes / limitations of this research which has been explained in the lines 13 to 83 on pages 2 and 3. 

Concern # 2: why a fuzzy logic has been employed? further motivations are needed

Author response: The revised manuscript is updated by including the discussion regarding Fuzzy logic Controller for information fusion in lines 84 to 91 on page 3 

Concern # 1:

Future works are limited. The author should further discuss this section. In the future, explainable AI aspects should be addressed in order to give a deeper understanding of the proposed system. In this sense the authors can refer to:

# A novel explainable machine learning approach for EEG-based brain-computer interface systems;

# Towards multi-modal causability with Graph Neural Networks enabling information fusion for explainable AI

Author response: Thank you so much. In the revised manuscript, the future work section has been rewritten which can be seen in lines 546 to 554 on page 18. Also, both relevant references have been included in the revised manuscript in the reference section on page 25

Comment # 2 Generally, this work is important and with the modifications this can be a really useful archival work for the international community. Congrats to the authors and this reviewer hopes that the suggestions help to improve the paper.

Author response: Thank you for your encouragement.

Reviewer # 3

-------------

Response to the Reviewers’ Concerns Reviewer # 3

Comment # 1: This manuscript presents a novelty ranking of mobile health applications. This ranking is based on the user's star ranking, users' text review, clinical approval, UI design, functionality, security, and privacy. Results suggest a competitive performance of the proposed ranking compared against app and expert rating.

Author response: Thank you very much.

Concern # 1: Review some misspelling words (invesgitation on page 2). I recommend reading the entire document to avoid this writing mistake.

Author response: Thank you for pointing out this issue. The manuscript has been thoroughly proofread to eliminate any typos and grammatical errors.

Concern # 2: Algorithm 1 is located several pages below its mention. I suggest including it below of its reference

Author response: Thank you. Algorithm 1 has now been placed on page 11 (line 384) and the references to the algorithm are mentioned afterwards.

Concern # 3: Into the manuscript are mentioned Questionaries Q1-Q4 for Clinical Approval measure. Please, include a detailed comment about this because it looks like a disconnected element. I understand that they are used for getting the clinical approval value, but the questionaries are not described. 

Author Response: Thank you. We have now included description of the Questions Q1-Q4 in the Approval and Certification section on page 13 (Lines 437--443).

Concern # 4: Are mentioned five scales for ranking apps: THESIS, MARS, Brief DISCERN, uMARS, and ORCHA-24. Compared with these, how is your proposal? is better, worst, or similar? Please provides a comparison with these methods.

Author response: Thank you for pointing out this issue. We have added the performance comparison of different scales in terms of Intra-class Correlation Coefficient (ICC). However, the performance of Brief DISCERN [50] could not be included in this comparison as it used a different evaluation scale. We updated the manuscript by including Figure 11 (D) on page 17, and the related description in lines 516-518 on the same page.

---

## [Decision Letter · Decision Letter 1]

9 Aug 2021

PONE-D-21-09300R1

User Reviews Influence Mobile Health Application Rating

PLOS ONE

Dear Dr. Kaiser,

Thank you for submitting your manuscript to PLOS ONE. After careful consideration, we feel that it has merit but does not fully meet PLOS ONE’s publication criteria as it currently stands. Therefore, we invite you to submit a revised version of the manuscript that addresses the points raised during the review process.

We look forward to receiving your revised manuscript.

Kind regards,

Junaid Rashid, Ph.D

Academic Editor

PLOS ONE

Journal Requirements:

Additional Editor Comments (if provided):

Thank you for improving the paper according to reviewers comments. There are some minor comments from the reviewer. Please incorporate these comments in the paper. In the paper use same terminology for figures and tables. Proofreading of the paper is required before final submission.

Reviewers' comments:

Reviewer's Responses to Questions

**Comments to the Author**

1. If the authors have adequately addressed your comments raised in a previous round of review and you feel that this manuscript is now acceptable for publication, you may indicate that here to bypass the “Comments to the Author” section, enter your conflict of interest statement in the “Confidential to Editor” section, and submit your "Accept" recommendation.

Reviewer #2: All comments have been addressed

Reviewer #3: All comments have been addressed

2. Is the manuscript technically sound, and do the data support the conclusions?

Reviewer #2: Yes

Reviewer #3: Yes

3. Has the statistical analysis been performed appropriately and rigorously? 

Reviewer #2: Yes

Reviewer #3: N/A

4. Have the authors made all data underlying the findings in their manuscript fully available?

Reviewer #2: Yes

Reviewer #3: Yes

5. Is the manuscript presented in an intelligible fashion and written in standard English?

Reviewer #2: Yes

Reviewer #3: Yes

6. Review Comments to the Author

Reviewer #2: The authors have been addressed all the comments accordingly.

I can now recommend this manuscript for publication.

Reviewer #3: The paper presents a significant improvement compared with the previous manuscript; however, I have some comments:

1. In Figure 11(D), a comparison among the scale against the proposal is detailed, but there is no discussion of what means these values concerning ICC. How is the proposal behavior based on this comparison?. Therefore, it is essential to mention it within the text.

2. In line 515, the authors use the "Fig." term, and the lines above use "Figure" please uses the same terminology in all documents.

7. PLOS authors have the option to publish the peer review history of their article (what does this mean?). If published, this will include your full peer review and any attached files.

Reviewer #2: No

Reviewer #3: No

---

## [Author Response · Author response to Decision Letter 1]

29 Aug 2021

Concern # 1: : In Figure 11(D), a comparison among the scale against the proposal is detailed, but there is no discussion of what means these values concerning ICC. How is the proposal behavior based on this comparison?. Therefore, it is essential to mention it within the text.

Author response: Thank you very much. In Figure 11 (D), a comparison among the scale has been presented. The discussion is now included in the revised manuscript.

Concern # 2: In line 515, the authors use the "Fig." term, and the lines above use "Figure" please uses the same terminology in all documents.

Author response: Thank you very much. We have replaced Fig. by Figures (on line 538) in the revised manuscript.

---

## [Editor Report · Decision Letter 2]

17 Sep 2021

ACCU3RATE: A Mobile Health Application Rating Scale Based on User Reviews

PONE-D-21-09300R2

Dear Dr. Kaiser,

We’re pleased to inform you that your manuscript has been judged scientifically suitable for publication and will be formally accepted for publication once it meets all outstanding technical requirements.

Kind regards,

Junaid Rashid, Ph.D

Academic Editor

PLOS ONE

---

## [Editor Report · Acceptance letter]

18 Nov 2021

PONE-D-21-09300R2 

ACCU^3^RATE: A Mobile Health Application Rating Scale Based on User Reviews 

Dear Dr. Kaiser:

I'm pleased to inform you that your manuscript has been deemed suitable for publication in PLOS ONE. Congratulations! Your manuscript is now with our production department. 

Kind regards, 

on behalf of

Dr. Junaid Rashid 

Academic Editor

PLOS ONE